# Comparing the Application Effects of Immersive and Non-Immersive Virtual Reality in Nursing Education: The Influence of Presence and Flow

**DOI:** 10.3390/nursrep15050149

**Published:** 2025-04-29

**Authors:** Choon-Hoon Hii, Cheng-Chia Yang

**Affiliations:** 1Department of Emergency Medicine, Kuang-Tien General Hospital, Taichung 433, Taiwan; choonhoonhii@gmail.com; 2Department of Healthcare Administration, Asia University, Taichung 413, Taiwan

**Keywords:** virtual reality, nursing education, flow, cognitive–affective model of immersive learning, cognitive absorption

## Abstract

**Background:** This study extends the theoretical framework based on the Cognitive–Affective Model of Immersive Learning (CAMIL) by incorporating flow state and cognitive absorption to investigate the effectiveness of virtual reality (VR) in nursing education. **Methods:** A randomized experimental design was adopted. A total of 209 students from three nursing assistant training centers in Taiwan were recruited through convenience sampling and randomly assigned to either immersive virtual reality (IVR) or Desktop VR groups for nasogastric tube feeding training. Data were collected through structured questionnaires and analyzed using partial least squares structural equation modeling (PLS-SEM). **Results:** The results revealed that immersion, curiosity, and control significantly impacted presence, which, in turn, positively influenced the flow state (β = 0.81, *p* < 0.001). Flow demonstrated positive effects on intrinsic motivation (β = 0.739, *p* < 0.001), situational interest (β = 0.742, *p* < 0.001), and self-efficacy (β = 0.658, *p* < 0.001) while negatively affecting extraneous cognitive load (β = −0.54, *p* < 0.001). Multigroup analysis showed that IVR had a stronger control–presence effect (|diff| = 0.337, *p* = 0.016), and flow had a great effect on motivation (|diff| = 0.251, *p* = 0.01), interest (|diff| = 0.174, *p* = 0.035), and self-efficacy (|diff| = 0.248, *p* = 0.015). Desktop VR more effectively reduced cognitive load (|diff| = 0.217, *p* = 0.041). **Conclusions:** These findings provide theoretical insights into the role of flow in VR learning and practical guidance for implementing VR technology in nursing education.

## 1. Introduction

Virtual reality (VR) is a three-dimensional simulation environment created using computer technology. It allows users to experience scenarios similar to the real world through visual, auditory, and other senses [1]. Recent studies have explored the effects of VR on learners’ knowledge, skills, attitudes, and behaviors [2,3,4,5,6]. In nursing and other health professions, VR learning is recognized as an effective teaching method that can provide a safe, authentic, diverse, and interactive learning environment, enhancing learning effectiveness and motivation [7,8]. It helps to improve nursing students’ learning satisfaction [8,9]. Moreover, it can effectively enhance the knowledge and skills of clinical professionals [10,11].

However, reference [12] indicated that there are shortcomings in the current research on VR learning, such as overlooking the unique characteristics of VR (e.g., immersion and interactivity), and a theoretical basis to explain how VR technology promotes learning effectiveness is also lacking. To address this gap, Makransky and Petersen [13] proposed the Cognitive–Affective Model of Immersive Learning (CAMIL)—a research-based theoretical framework that explains how learning occurs in immersive environments. Although CAMIL is not technology-specific and applies to immersive learning technologies in general, it builds on a recent wave of media comparison studies involving VR. This framework takes a constructivist view of learning, emphasizing the active construction of knowledge through immersive experiences rather than the passive acquisition of information. It provides a structured approach to understanding the complex relationships between technological features, psychological experiences, and learning outcomes in virtual environments. According to CAMIL, the technological features of immersion and interactivity in VR influence presence. Presence influences both affective and cognitive mechanisms, including situational interest, intrinsic motivation, self-efficacy, and cognitive load. These affective and cognitive mechanisms influence learning outcomes.

In terms of VR, presence is generally defined as the degree to which a person’s feelings are transformed from the real environment into the virtual environment; it is the subjective feelings of learners in the virtual environment [14]. This study suggests that presence does not necessarily directly affect the cognitive and emotional aspects of learning. Instead, the interactions, perceptions, and situations in the virtual environment may cause these changes. Several studies have found a close relationship between presence and flow [15,16,17]. Flow is a psychological state in which an individual is fully engaged in an activity while enjoying the experience of the present moment; flow involves the feeling that time is passing quickly while ignoring or forgetting information unrelated to the task [18]. A high level of immersion and interaction in VR can create an immersive sense of presence, generating a flow state [19].

Flow often occurs during tasks that are highly engaging and meaningful. It may boost their intrinsic motivation and self-efficacy when individuals believe that the difficulty of a task matches their ability and that they can overcome the challenges to complete it [20,21]. Furthermore, when task difficulty suits a learner’s ability, it can stimulate their initiative and generate situational interest [22]. A flow state is characterized by a high degree of concentration and deep engagement. In this state, an individual’s attention is focused on the current task, enabling them to process information effectively while reducing the influence of the external cognitive load. Flow can enhance learner engagement in VR, leading to a more effective learning experience. Therefore, flow may be a key factor in the VR learning process.

Based on CAMIL, the antecedents that affect VR presence may be immersion and interactivity [13]. However, in addition to using “immersion” and “interactivity” to explain presence, this study argues that the antecedents affecting presence can be discussed through the concepts of cognitive absorption [23]. Cognitive absorption is when users interact with software and information technology. It has been used to understand individual evaluations of virtual technology [24,25]. In recent years, virtual reality systems have been categorized according to their level of immersion into immersive VR (IVR) [26], which typically uses head-mounted displays, and non-immersive or Desktop VR [27], which uses standard screens and conventional input devices. While both types have been used in nursing education [2,7,8,11], existing studies have mainly focused on comparing VR to traditional or video-based instructions. However, the isolated impact of immersion level, independent of other instructional variables, has not been sufficiently elucidated in the existing literature. Thus, comparing IVR and Desktop VR provides a unique opportunity to examine whether an increased sense of immersion translates into greater presence, deeper flow experiences, and improved learning outcomes. Based on the CAMIL architecture, this study explored the impact of different types of VR on learning outcomes (Desktop VR and IVR). The purpose of this study is as follows: (1) to examine the antecedent factors that affect VR presence; (2) to verify the relationship between VR presence and flow; (3) to discuss the relationship between flow, learning emotion, and cognitive factors; and (4) to compare the learning models of different immersion levels (IVR and Desktop VR).

## 2. Theoretical Background and Hypotheses

### 2.1. Antecedent Factors Affecting Presence in VR

Cognitive absorption focuses on the user’s experience of interacting with information technology [23], which encompasses five dimensions: (1) temporal dissociation—ignoring the flow of time during the human–technology interaction; (2) focused immersion—people’s interaction with a task without any interference from external factors, similar to the concept of immersion; (3) heightened enjoyment—the positive experience brought about by the interaction between individuals and activities; (4) curiosity—cognitive and sensory curiosity stimulated by technology; and (5) control—whether people can fully grasp the results of their interaction with tasks, similar to the concept of interactivity. Cognitive absorption can be used to understand the individual assessments of virtual technology use [24,25]. Research indicates that these dimensions significantly impact presence [28,29].

Therefore, we posit that learners’ sense of presence in the virtual environment may be enhanced through several experiential aspects of VR interaction. These include the following: experiencing temporal dissociation (becoming so engaged that they lose track of time); achieving focused immersion (concentrating on the task while ignoring external distractions); feeling heightened enjoyment during the learning process; exerting control by consciously mastering task execution; and engaging curiosity, which drives imaginative involvement throughout the experience. Based on these relationships, we propose the following hypothesis:

**H1:** 
*The experiential process of VR, including (a) temporal dissociation, (b) focused immersion, (c) heightened enjoyment, (d) curiosity, and (e) control, positively impacts presence.*


### 2.2. The Relationship Between Presence and Flow in VR

Presence in VR refers to the degree of immersion a person experiences when transitioning from a real to a virtual environment [14]. Presence can be defined by two modes: descriptive and structural. The former emphasizes the constituent elements of VR presence, including spatiality and authenticity; the latter explains how VR generates presence in the brain [30]. Flow refers to excluding or forgetting information irrelevant to the task when one is focused on the present activity. The flow argument represents a state that arises from the assumption of interaction between individual and situational factors [31]. Flow occurs when three main conditions are met: (1) clear goals, meaning that the person in flow is intensely aware of the goals and actions they want to achieve; (2) unambiguous feedback through the guidance of clear objectives, individuals obtain real-time feedback from activities; and (3) challenge–skill balance: it refers to the challenge posed by the activity. This must match the individual’s ability to generate flow.

Reference [16] suggests that presence and flow are different concepts entirely, but they may influence each other. For instance, presence describes the process of being immersed in the virtual world, while flow refers to the experience of being immersed in a given task. The more users are immersed in the virtual experience, the more likely they are to enter a flow state when completing a task [24,32]. Based on the above facts, when learners focus on immersive VR learning tasks, they can feel a sense of being in the scene. This experiential process is consistent with the three conditions of flow: (1) through clear instructions in VR tasks, learners have clear goals of what they want to achieve; (2) through the interactive process, learners receive unambiguous real-time feedback on their performance; and (3) the VR environment allows tasks to be designed at an appropriate level of challenge that matches learners’ abilities. When these conditions are met, learners experience satisfaction from task completion, leading to a pleasant emotional state that motivates them to continue engaging in the learning activities, representing the flow state. Thus, we propose the following hypothesis:

**H2:** 
*Presence positively impacts flow.*


### 2.3. Flow and Intrinsic Motivation

Intrinsic motivation is the tendency to actively engage in a task because one finds it interesting, challenging, or enjoyable [33]. Elements of intrinsic motivation [34] include autonomy (individuals feel that they can take charge and control their own behavior), individual competence (how well individuals perform tasks), and clarity of purpose (clearly setting goals and being able to obtain real-time feedback on their progress). These elements share similarities with the characteristics of flow. Flow often occurs when a challenge is targeted while the task matches an individual’s abilities. Individuals are more likely to experience a state of flow when they are competent enough to handle a particular task and receive feedback from the process of performing it. However, intrinsic motivation and flow differ. Intrinsic motivation is associated with a task because it is intrinsically satisfying [33]. Flow occurs automatically after participating in an activity because an individual feels that the activity is worthwhile [31]. Past studies have confirmed the relationship between flow and intrinsic motivation [21,35,36]. Thus, we suggest that the characteristics of VR teaching can produce a presence for users; presence enables learners to immerse themselves in the experience of a certain task and enter a flow state. During the flow state, a learner feels that the difficulty of the task is suitable for their ability while receiving feedback in real time during the challenge. This results in more intrinsic motivation to devote to the task, illustrating why people in a flow state are more focused and enjoy their learning activities [16]. Therefore, we propose the following hypothesis:

**H3:** 
*Flow positively impacts intrinsic motivation.*


### 2.4. Flow and Situational Interest

Situational interest refers to an individual’s sense of interest stimulated by the situational environment [37]. Therefore, situational interest is spontaneous, transient, and stimulated by the external environment [38]. Research suggests that to stimulate people’s situational interest, interactive teaching content should be designed while considering the balance of challenging tasks and students’ abilities in mind, focusing on clear teaching objectives and real-time feedback with improved autonomy and entertainment [22]. These design elements are all characteristics of flow [18]. Therefore, the relationship between flow and situational interest can be mutually influential [22,39]. This may explain why creating fun and engaging learning environments is essential to improving student engagement and learning outcomes. Based on the above facts, we suggest that VR teaching inspires a state of flow that exhibits novelty and optimal challenges, creating more learning intention. This can strengthen the situational interest of learners. Therefore, we propose the following hypothesis:

**H4:** 
*Flow positively impacts situational interest.*


### 2.5. Flow and Self-Efficacy

Self-efficacy refers to an individual’s ability to have confidence in the expected outcomes of performing a task [40]. This is the degree to which an individual can use their ability, believe they can do something, and achieve a goal. People with high self-efficacy are capable of prescience and self-reflection [40]. They focus on sticking to their goals to achieve their desired results. Self-efficacy can change over time, depending on an individual’s experience. Such experiential reinforcement is often created by flow; thus, it can be assumed that the process of acquiring flow increases self-efficacy [40]. Furthermore, flow is characterized by a strong sense of control over a task [18]; this sense of control helps to master upcoming challenges and increase self-confidence, similar to self-efficacy. Research has confirmed a positive relationship between flow and self-efficacy [41]. Therefore, when learners are in a flow state, it helps them to exhibit higher abilities in balancing challenges and skills, goal setting, concentration, and task control (i.e., they are in a high flow state). It also helps to enhance their confidence in achieving expected results in the face of tasks (i.e., self-efficacy). Therefore, we propose the following hypothesis:

**H5:** 
*Flow positively impacts self-efficacy.*


### 2.6. Flow and the Extraneous Cognitive Load

Cognitive load is a learning load that occurs when the amount of information or the way it is presented exceeds a person’s working memory-carrying capacity [42]. Improvement through different learning tools, e.g., multimedia visual effects and audio methods, to process learning information can reduce the load on working memory while enhancing learning effectiveness [43]. Several VR-related studies have found that the extraneous cognitive load is essential for understanding the VR learning process. For instance, the fidelity of VR images, complex visual design, detailed images, and more environmental interaction effects can produce an extraneous cognitive load unrelated to learning [5,8,44,45]. Therefore, the cognitive load discussed in this study refers only to the extraneous cognitive load. Reference [46] confirms that flow reduces the impact of the extraneous cognitive load and improves problem-solving ability. The flow created through instructional design or interactive elements can reduce students’ exposure to poor teaching materials or methods. Based on the above facts, we suggest that when VR teaching allows people to be in a state of flow, it helps to reduce the impact on their extraneous cognitive load because they are focused on the current task situation. Therefore, we propose the following hypothesis:

**H6:** 
*Flow negatively impacts the extraneous cognitive load.*


### 2.7. Adjustment Effects of Different VR Types

Studies indicate that highly immersive VR learning elicits more positive outcomes for learning than non-immersive VR learning [4,5,6,47]. It can be understood that a learning system with high immersion can lead to better presence, while presence can positively affect learning emotions and improve perceived learning value [48]. This study focused on comparing the virtual environments of Desktop VR and IVR. The difference between Desktop VR and IVR is indicated by the degree of immersion. Regarding the experiential process, we posit that the virtual environment constructed through IVR may be more effective than that constructed through Desktop VR. For example, IVR may significantly enhance a user’s sense of immersion, control, and temporal dissociation through more intense visual stimulation and unique head-mounted display operation, significantly enhancing the sense of presence. Thus, IVR may be more effective than Desktop VR in inducing a flow experience through presence, thereby enhancing intrinsic motivation and situational interest and improving self-efficacy. Thus, we propose the following hypotheses:

**H7:** 
*Different types of VR (Desktop VR and IVR) can regulate the impact of the VR experience process on presence, such as (a) temporal dissociation, (b) focused immersion, (c) heightened enjoyment, (d) curiosity, and (e) control.*


**H8:** 
*Different types of VR (Desktop VR and IVR) can regulate the impact of presence on flow.*


**H9:** 
*Different types of VR (Desktop VR and IVR) can regulate the impact of flow on (a) intrinsic motivation, (b) situational interest, (c) self-efficacy, and (d) temporal cognitive load.*


## 3. Materials and Methods

### 3.1. Conditions for Sample Collection

This study adopted an experimental design with random assignment to compare IVR and Desktop VR environments and to examine their effects on key constructs within the CAMIL framework, including presence, flow, and learning-related outcomes. Participants were recruited through convenience sampling from nursing assistant training courses at three educational and training institutions in central Taiwan. Data collection started in October 2022. Inclusion criteria were as follows: (1) adults aged 20 years or older; (2) those who had never participated in similar nursing training programs; and (3) those without a history of neurological, psychiatric, or cardiovascular disorders or cognitive impairment.

Eligible participants who provided informed consent were randomly assigned to either the immersive virtual reality (IVR) group or the Desktop VR group in a 1:1 ratio using a computer-generated randomization sequence. A total of 220 participants were initially enrolled. During the intervention, eight participants withdrew for health reasons (vertigo, COVID-19, or eye discomfort), and three withdrew for personal reasons, leaving 209 valid participants (105 in the Desktop VR group and 104 in the IVR group). The required sample size was calculated using G*Power 3.1.9.7 [49], based on a minimum R^2^ value of 0.15 and 11 predictor variables, with a statistical power level of 0.71 [8]. The final sample size exceeded this threshold, ensuring adequate power for the analyses.

### 3.2. Experimental Content and Research Steps

The instructional content in this study was selected and structured to align with both clinical practice standards and the theoretical constructs of the Cognitive–Affective Model of Immersive Learning (CAMIL). CAMIL posits that immersion and interactivity enhance a sense of presence, thereby triggering psychological processes such as emotional engagement, intrinsic motivation, situational interest, and cognitive processing. The nasogastric tube feeding scenario was chosen for its high contextual relevance and potential to elicit immersive engagement. By simulating realistic and sequential clinical tasks—including preparation, patient interaction, feeding procedures, and follow-up care—the content was designed to maximize presence and facilitate flow within a clinically meaningful learning experience.

The intervention was implemented using a previously validated instructional framework for nasogastric tube feeding training [8], developed in accordance with the national clinical care guidelines and standard operating procedures. Prior to the VR intervention, all participants received a 50- min theoretical course consisting of a 30 min lecture and a 20 min group discussion to ensure a basic understanding of the procedure. After the theoretical session, participants proceeded to the simulation-based training using either IVR or Desktop VR. Both groups received the same instructional content, including preparation steps, meal setup, feeding, post-care cleaning, and medication assistance.

The virtual learning environments were built in Unity 3D and programmed in C#. Relevant 3D models were created in 3ds Max and integrated into the system. All content was standardized to ensure consistency between groups, with the only difference being the level of immersion provided by the VR modality. In the IVR group, participants used HTC VIVE Focus Plus head-mounted displays and handheld controllers. They navigated the virtual environment using a joystick and interacted with objects using the controller buttons. The Desktop VR group used a computer, a mouse, and a keyboard. Navigation was performed using the arrow keys and mouse movements, while object interactions were performed using mouse clicks. Participants completed the simulation at their own pace. On average, the IVR group spent 26.3 min completing the tasks, while the Desktop VR group completed their tasks in approximately 20.5 min. Upon the completion of VR training, all participants immediately completed the structured questionnaire.

### 3.3. Research Measurement Tools

The questionnaire was primarily a self-report instrument and consisted of five parts. This study used a 5-point Likert scale ranging from 1 point (strongly disagree) to 5 points (strongly agree) for measurement. The first part included the basic data on age, gender, education level, and marital status. The second part assessed cognitive absorption based on the framework proposed by Agarwal and Karahanna [23], who defined it as “a state of deep involvement with software”. It included 15 items across five subdimensions: temporal dissociation, focused immersion, heightened enjoyment, control, and curiosity. Each subscale consisted of three items adapted to the VR caregiving learning context. Previous studies have reported acceptable-to-high internal consistency for these subdimensions, with Cronbach’s alpha values ranging from 0.67 to 0.87 [28].

The third part used Vorderer et al.’s [50] presence scale, which conceptualizes presence as the feeling of “being there”. This eight-item scale measures participants’ perceptions of spatial presence, self-location, and involvement in the virtual environment. The fourth section adapted four items from Kim and Hall [9] to measure flow states during VR learning, capturing essential components such as concentration, enjoyment, control, and the merger of action and awareness. Both the presence and flow scales have demonstrated strong internal consistency in a previous study, with reported Cronbach’s alpha values ranging from 0.864 to 0.926 [29].

The fifth part assessed learners’ motivational and cognitive–affective responses, which are essential for understanding engagement and perceived task difficulty in VR-based learning. It included four items on intrinsic motivation [51], which measure the learner’s internal drive and enjoyment; five items on situational interest [51,52], which reflect interest and attention triggered by the learning content (e.g., “The caregiving tasks in VR were interesting”); four items on self-efficacy [51], which assess the learner’s perceived confidence in completing caregiving tasks; and three items on extraneous cognitive load [53], assessing the perceived mental effort required, which have demonstrated acceptable internal consistency in previous studies (Cronbach’s alpha typically above 0.70). For transparency and reproducibility, the full set of questionnaire items is included in Appendix A.

### 3.4. Data Analysis

This study adopted a partial least squares (PLS) method to construct predictive models. For causal model analysis between potential variables, this method is superior to the general linear structural relationship model and is suitable for exploratory research. It applies not only to the dimension of a single item but also has predictive and explanatory power. It is not limited by the variable distribution pattern or the number of samples and is suitable for multigroup analysis (MGA) [54]. These attributes helped to explore the differences in learning models between VR types. We used the Smart PLS software (version 4.1.1.1) to analyze the measurement model and structure and selected 5000 samples using Bootstrap resampling for parameter calculation and inference estimation.

## 4. Results

### 4.1. Sample Descriptive Statistics

Demographic characteristics of the participants are presented in Table 1. Among the 209 participants, 79 (37.8%) were 41–50 years old, and 60 (28.7%) were 51–60 years old. The youngest participant was 22 years old. The oldest participant was 60 years old. The average age of participants was 38.74 years old. There were more women participants than men participants, accounting for 55.98%. Most participants were high school graduates (63.6%), followed by university graduates (36.4%).

### 4.2. Reliability and Validity of the Research Instruments

This study used individual item reliability, composite reliability (CR) of potential variables, and average variance extracted (AVE) of potential variables to evaluate the measurement model. Among them, “intrinsic motivation” and “situational interest” each had a factor loading of less than 0.5 and were deleted. The factor loading coefficients of the other variables ranged from 0.719 to 0.948, all higher than the recommended value of 0.5. The CR values of each variable ranged from 0.806 to 0.961, above the standard of 0.7, indicating that the research model has good internal consistency. The AVE values ranged from 0.572 to 0.892, higher than the standard values of 0.5 (See Table 2).

Discriminant validity is obtained by checking that the square root of the average variance extracted (AVE) for each construct exceeds its correlation coefficients with other constructs. Overall, the results indicate that all dimensions have satisfactory levels of reliability, convergent validity, and discriminant validity (see Table 3). Therefore, the variables in this study have good convergence validity. We also used the heterotrait–monotrait ratio (HTMT) to evaluate discriminant validity. All HTMT ratios were below the threshold value of 0.85, confirming discriminant validity. The results of the standardized root mean square residual (SRMR) were used as a goodness-of-fit measure based on PLS-SEM. The complete dataset had a value of 0.054, indicating that all data satisfied the requirements for goodness-of-fit. The VIF values for all indicators/manifested variables were between 1.088 and 4.864; the VIF value did not exceed the threshold value of 5, indicating no multicollinearity issues.

### 4.3. Hypothesis Verification

We used Bootstrap resampling in PLS to test the structural model’s path significance. R^2^ is the main indicator for determining the quality of a model [55]. We used PLS to estimate the path relationships between various dimensions; the path values were standardized coefficients to verify the hypothesis regarding the path relationships in the research model. All relevant hypotheses must reach a significance level of α = 0.05 to be valid (see Table 4).

The results indicate that immersion, curiosity, and control all led to significant differences in presence (β = 0.146, t = 2.115, *p* = 0.035; β = 0.281, t = 2.418, *p* = 0.006; β = 0.367, t = 5.722, *p* < 0.001). This indicates that the immersion, curiosity, and level of control elicited by the experiential process of VR resulted in a stronger sense of presence. Therefore, H1b, H1d, and H1e were established. This study did not find a significant effect of temporal dissociation and enjoyment on presence. The above three factors could explain 78.6% of the variance in terms of presence. The presence generated by VR positively affected the flow state (β = 0.81, t = 29.07, *p* < 0.001); the higher the presence, the more likely it was to produce a flow state. Therefore, H2 is established, and 65.1% of the flow variance could be explained. We also found that the stronger the flow feeling, the stronger the intrinsic motivation, situational interest, and self-efficacy (β = 0.739, t = 16.62, *p* < 0.001; β = 0.742, t = 19.78, *p* < 0.001; β = 0.658, t = 14.07, *p* < 0.001). Furthermore, we found that a flow state can reduce the extraneous cognitive load of individual learning (β = −0.54, t = 9.90, *p* < 0.001), validating H3, H4, H5, and H6 (see Figure 1).

### 4.4. Multiple Group Analysis

To test H7, H8, and H9, we divided all participants into two groups (IVR and Desktop VR). We conducted multiple PLS analyses to compare the differences in path relationships between the two groups. PLS has been widely adopted in past research [56]. In the multigroup analysis, the permutation test revealed no significant differences in several paths related to presence. Specifically, temporal dissociation did not show a significant difference in its effect on the presence (|diff| = 0.044, *p* = 0.729), leading to the rejection of H7a. Similarly, focused immersion (|diff| = 0.001, *p* = 0.992) and heightened enjoyment (|diff| = 0.254, *p* = 0.344) showed no significant differences, resulting in the rejection of H7b and H7c, respectively. Although curiosity yielded a relatively large coefficient difference (|diff| = 0.299), the result was not statistically significant (*p* = 0.204), and thus, H7d was also rejected. However, the relationship between control and presence showed a significant difference between groups (|diff| = 0.337, *p* = 0.016), with the IVR group exhibiting a stronger effect. Therefore, H7e was supported (see Table 5).

Presence did not exhibit a significant difference in flow (|diff| = 0.091, *p* < 0.05). Therefore, H8 was rejected. Additionally, flow exhibited significant differences in intrinsic motivation, situational interest, self-efficacy, and the extraneous cognitive load (|diff| = 0.251, *p* < 0.05; |diff| = 0.174, *p* < 0.05; |diff| = 0.248, *p* < 0.005; |diff| = 0.217, *p* < 0.05). IVR had a higher effect on intrinsic motivation, situational interest, and self-efficacy in flow compared to Desktop VR. However, compared to IVR, Desktop VR effectively reduced the extraneous cognitive load. The findings indicate that different types of VR had a moderating effect on control, presence, flow, and the relationship between learning emotions and cognition. Therefore, H9a–H9d are established (see Table 5).

## 5. Discussion

This study proposed a VR learning model based on the CAMIL framework. We developed a VR teaching system to enhance the skills of nasogastric tube feeding in nursing care. We discussed antecedents affecting VR presence while exploring the relationship between VR presence and flow. The influence of flow on learning, emotion, and cognition was also examined. The findings indicate that immersion, control, and curiosity significantly influenced presence (H1b, H1d, H1e, supported); a stronger sense of presence led to greater involvement in the experience, which, in turn, enhanced flow (H2, supported). Flow positively affected intrinsic motivation (H3, supported), situational interest (H4, supported), and self-efficacy (H6, supported) while reducing the extraneous cognitive load (H5, supported). Multigroup analysis further confirmed the moderating effects of immersion level, supporting H9a–H9d. The following description outlines this study’s empirical contributions.

### 5.1. Theoretical Significance

First, this study found that a sense of control was the most significant factor in predicting presence, aligning with the results from past research [13,28,29]. This means that the stronger the people’s sense of control when interacting with virtual environments, the stronger their sense of presence in VR. Second, users with higher curiosity tend to perceive a higher sense of presence in VR [28,57]. Perhaps due to VR’s innovative technology, users feel curious during their experience, further enhancing their presence. Finally, this study found that the stronger the immersion, the higher the degree of presence, which is consistent with reference [13]’s findings regarding the impact of VR presence on immersion.

Unlike previous studies, the model in this study introduces the concept of “flow” to explore how flow triggers learning emotions and cognitive responses in users. To the best of our knowledge, this study is the first to propose this architecture. While previous studies have indicated that high presence contributes to positive learning emotions [26,48,58,59], this study suggests that the learning process depends on the participation and autonomy of individuals, a learning concept of internal development constructivism [60]. When people participate in clinical teaching tasks through VR, immersion enables students to maintain focus and generate a flow state. Flow promotes learning interest, intrinsic motivation, and self-efficacy. Previous studies have examined isolated emotional factors, such as intrinsic motivation, situational interest, or self-efficacy, in relation to flow independently [22,35,41]. The uniqueness of this study lies in the comprehensive consideration of the learning emotional effects based on the CAMIL framework, which adds value to establishing VR learning models.

Second, although past studies have indicated that VR’s high fidelity and complex visual design tend to trigger an extraneous cognitive load unrelated to learning [5,8,44], this study emphasizes that VR presence can induce learners’ flow state, thus affecting the extraneous cognitive load. During a flow state, learners’ concentration is improved, and less attention is paid to information unrelated to learning [46,61,62]. Therefore, if VR design content is attractive enough—with a sense of presence while aligning with the interests and abilities of learners—it is easier to trigger flow and immerse learners, becoming a key factor in reducing the extraneous cognitive load.

Finally, this study tested VR systems with differing levels of immersion (IVR, Desktop VR) to validate the model’s results. First, IVR enables people to immerse themselves in tasks through a VR headset and interact with the virtual environment through a VR handle. Comparatively, Desktop VR only involves mouse control. Thus, IVR provides a higher level of control and enhances the sense of presence. Second, learning IVR can lead to higher self-efficacy and situational interest. This finding aligns with the results of previous studies [8,63,64]. Notably, although previous studies have found that IVR reduces the extraneous cognitive load [65], this may be due to the flow state created by VR. Our study confirms that flow in IVR reduces the extraneous cognitive load, but Desktop VR has a greater effect in reducing this load compared to IVR. This difference may be because IVR requires users to manipulate handlebars and HMDs, which are more complex than Desktop VR training environments [8]. Therefore, Desktop VR appears more effective in reducing the extraneous cognitive load.

### 5.2. Practical Significance

Based on these findings, this study provides practical suggestions and guidelines for VR teaching systems. Since a sense of control can affect presence, instructional designers can focus on creating instructional content that allows users to interact with scenes, thereby increasing their sense of control. For example, systems can be designed to enable users to move and use virtual objects, thereby increasing presence. This study also suggests that the influence of curiosity on presence should not be overlooked. Previous studies indicate that VR may be a tool to encourage curiosity [66], and curiosity indeed affects presence in VR [28,67]. Therefore, the advantages of VR can be used to create rich visual and auditory experiences and to design fascinating situations and task challenges. This can help stimulate users’ curiosity and further influence their presence.

Presence generated through VR is an essential factor in promoting a user’s flow, and flow is the antecedent to creating learning emotion. Therefore, to improve the flow state of learners, we suggest adding challenging tasks to VR learning materials. This teaching method, similar to game design, can be used to design challenging learning tasks through powerful narrative content to stimulate learners’ continual participation and challenges. This helps to create a flow state [19], which will stimulate learning motivation, situational interest, and self-efficacy while reducing the influence of the extraneous cognitive load. Teachers can flexibly choose to use IVR or Desktop VR based on their teaching objectives, student needs, and the immersion requirements of the course. Specifically, for situations requiring the cultivation of clinical reasoning and situational response, IVR may be more effective; for situations that reduce the learning load and cultivate basic knowledge, Desktop VR may be more suitable.

### 5.3. Limitations and Suggestions

The study limitations are as follows. First, the participants were only recruited from three nursing assistant training institutions. Although the participants’ age distribution was wide, we suggest recruiting more participants for future studies to confirm the validity of the research framework. Moreover, this study focused on one specific skill (the nasogastric tube feeding technique); other care skills should be explored in subsequent studies. Second, this study aimed to verify the effect of presence on the state of flow rather than the “process of flow”. We measured the learner’s transient flow experience. Therefore, the duration of the participants’ flow should be further explored. Future studies should also measure flow in a way that can reflect the flow process [68] to more fully understand the changes that occur during the learner’s experience. In terms of methodological considerations, although the randomized experimental design helped control potential confounding variables and improve internal validity, the use of convenience sampling may limit the generalizability of the findings. Additionally, this study focused on the immediate effects of VR-based learning within a single session. Although flow and other psychological outcomes were captured directly after the intervention, it remains unclear whether these effects are sustained over time. Future studies could adopt a longitudinal design to investigate the persistence of learning motivation, interest, and self-efficacy.

## 6. Conclusions

This study developed a theory-based VR learning framework for nursing skill training, with a specific focus on nasogastric tube feeding techniques. It found that cognitive absorption—specifically immersion, curiosity, and control—plays a significant role in shaping presence in VR learning environments. This study further explored the strong relationship between presence and flow and examined how flow positively influenced learning emotions and cognitive factors. Empirical results confirmed the moderating effect of VR learning systems with different levels of immersion on learning outcomes. Specifically, IVR proved to be more effective than Desktop VR in enhancing flow, learning emotions, and cognitive engagement, while Desktop VR was better at reducing the extraneous cognitive load. These findings provide both a theoretical foundation and practical guidance for the design of effective VR-based learning systems.

## Figures and Tables

**Figure 1 nursrep-15-00149-f001:**
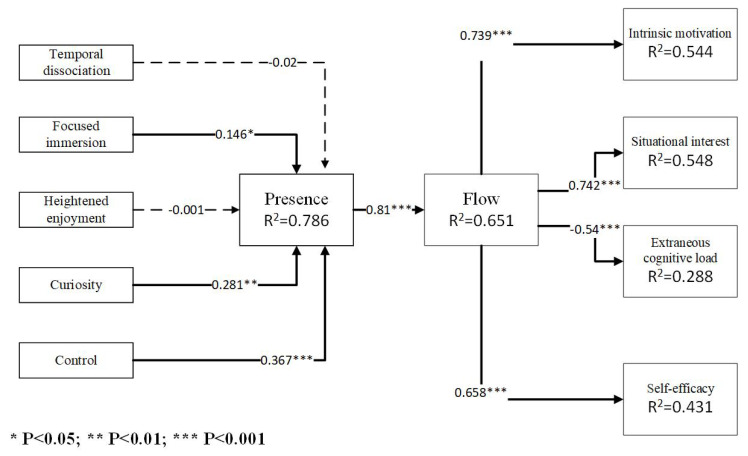
Structural analysis of the research model.

**Table 1 nursrep-15-00149-t001:** Demographic characteristics of study participants (N = 209).

Variable	Category	Total (%)	IVR (%) N = 104	Desktop VR (%) N = 105
Gender	Female	117 (55.98%)	61 (58.7%)	56 (53.3%)
	Male	92 (44.02%)	56 (53.8%)	48 (45.7%)
Age	21–30 years	35 (16.75%)	19 (54.3%)	16 (45.7%)
	31–40 years	35 (16.75%)	20 (57.1%)	15 (42.9%)
	41–50 years	79 (37.80%)	43 (54.4%)	36 (45.6%)
	51–60 years	60 (28.70%)	36 (60.0%)	24 (40.0%)
Education level	High school	133 (63.6%)	74 (55.6%)	59 (44.4%)
	University	76 (36.4%)	45 (59.2%)	31 (40.8%)
Marital status	Single	80 (38.28%)	45 (56.2%)	35 (43.8%)
	Married	124 (59.33%)	72 (58.1%)	52 (41.9%)
	Divorced	2 (0.96%)	1 (50.0%)	1 (50.0%)
	Other (e.g., cohabiting, separated)	3 (1.43%)	2 (66.7%)	1 (33.3%)

**Table 2 nursrep-15-00149-t002:** Validity and reliability.

Dimension	Variable	Loading	T-Value	Cronbach’s Alpha	CR	AVE
Temporal dissociation	Tim 1	0.916	43.561	0.822	0.918	0.849
	Tim 2	0.875	20.684			
	Tim 3	0.913	43.167			
Focused immersion	Imm 1	0.868	20.358	0.866	0.908	0.713
	Imm 2	0.846	28.457			
	Imm 3	0.877	31.679			
Heightened enjoyment	Fun 1	0.929	67.405	0.955	0.961	0.892
	Fun 2	0.95	102.111			
	Fun 3	0.955	103.634			
Control	Con 1	0.863	34.107	0.862	0.915	0.783
	Con 2	0.878	37.584			
	Con 3	0.913	57.711			
Curiosity	Cur 1	0.95	82.018	0.937	0.96	0.888
	Cur 2	0.948	74.064			
	Cur 3	0.93	57.22			
Presence	Pre 1	0.883	42.338	0.957	0.963	0.769
	Pre 2	0.874	40.577			
	Pre 3	0.893	44.649			
	Pre 4	0.926	66.873			
	Pre 5	0.879	35.691			
	Pre 6	0.892	62.675			
	Pre 7	0.893	47.052			
	Pre 8	0.768	19.840			
Flow	Flow 1	0.879	46.454	0.868	0.912	0.718
	Flow 2	0.872	49.595			
	Flow 3	0.888	53.655			
	Flow 4	0.742	14.558			
Intrinsic motivation	Imt 1	0.856	42.160	0.908	0.935	0.783
	Imt 2	0.891	42.427			
	Imt 3	0.907	47.707			
	Imt 4	0.886	39.375			
Situational interest	Si 1	0.926	82.358	0.946	0.937	0.751
	Si 2	0.941	77.180			
	Si 3	0.935	70.133			
	Si 4	0.873	23.461			
	Si 5	0.746	18.96			
Extraneous cognitive load	Cl 1	0.767	11.560	0.806	0.800	0.572
	Cl 2	0.719	5.518			
	Cl 3	0.782	10.293			
Self-efficacy	Sf 1	0.882	43.803	0.920	0.932	0.774
	Sf 2	0.854	32.992			
	Sf 3	0.907	54.293			
	Sf 4	0.876	41.864			

**Table 3 nursrep-15-00149-t003:** Discriminant validity.

	1	2	3	4	5	6	7	8	9	10	11
1. Temporal dissociation	**0.921**										
2. Focused immersion	0.644	**0.872**									
3. Heightened enjoyment	0.688	0.717	**0.954**								
4. Control	0.61	0.658	0.752	**0.885**							
5. Curiosity	0.616	0.804	0.794	0.804	**0.942**						
6. Presence	0.592	0.687	0.757	0.839	0.815	**0.877**					
7. Flow	0.654	0.638	0.731	0.75	0.72	0.808	**0.847**				
8. Intrinsic Motivation	0.675	0.665	0.822	0.811	0.791	0.733	0.739	**0.885**			
9. Situational Interest	0.708	0.682	0.81	0.802	0.777	0.745	0.742	0.812	**0.928**		
10. Self-efficacy	0.521	0.596	0.685	0.716	0.657	0.733	0.658	0.662	0.749	**0.891**	
11. Extraneous Cognitive Load	−0.526	−0.486	−0.569	−0.593	−0.575	−0.565	−0.54	−0.609	−0.64	−0.604	**0.727**

Note: The square root of the AVE values is shown in bold. Off-diagonal elements are the inter-construct correlations.

**Table 4 nursrep-15-00149-t004:** Results of the research hypotheses.

Hypothesis	Path Relation	Path Coefficient	T-Value	Result
H1a	Temporal Dissociation → Presence	−0.02	0.286	False
H1b	Focused Immersion → Presence	0.146	2.115 *	True
H1c	Heightened Enjoyment → Presence	−0001	0.014	False
H1d	Curiosity → Presence	0.281	2.418 *	True
H1e	Control → Presence	0.367	5.722 ***	True
H2	Presence → Flow	0.81	29.07 ***	True
H3	Flow → Intrinsic Motivation	0.739	16.62 ***	True
H4	Flow → Situational Interest	0.742	19.78 ***	True
H5	Flow → Extraneous Cognitive Load	0.54	9.90 ***	True
H6	Flow → Self-Efficacy	0.658	14.07 ***	True

*: *p* < 0.05; ***: *p* < 0.001.

**Table 5 nursrep-15-00149-t005:** Multigroup comparison test results.

Relationship	Path Coefficient (IVR)	Path Coefficient (Desktop VR)	Path Coefficient Difference	*p*-Value	Significant Difference?	Hypothesis
Temporal dissociation-->Presence	0.006	0.05	−0.044	0.729	No	H7a
Focused immersion-->Presence	0.119	0.12	−0.001	0.992	No	H7b
Heightened enjoyment-->Presence	0.163	−0.091	0.254	0.344	No	H7c
Curiosity-->Presence	0.481	0.182	0.299	0.204	No	H7d
Control-->Presence	0.458	0.121	0.337	0.016	Yes	H7e
Presence-->Flow	0.859	0.768	0.091	0.122	No	H8
Flow-->Intrinsic Motivation	0.892	0.641	0.251	0.010	Yes	H9a
Flow-->Situational Interest	0.853	0.679	0.174	0.035	Yes	H9b
Flow-->Self-efficacy	0.798	0.55	0.248	0.015	Yes	H9c
Flow-->Extraneous Cognitive Load	−0.468	−0.685	0.217	0.041	Yes	H9d

## Data Availability

The data presented in this study are available upon request from the corresponding author due to ethical restrictions imposed by the Institutional Review Board (IRB).

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
