# Peer review of "Comparing the Application Effects of Immersive and Non-Immersive Virtual Reality in Nursing Education: The Influence of Presence and Flow"

_nursrep, 2025, doi:10.3390/nursrep15050149_

Round 1
Reviewer 1 Report
Comments and Suggestions for Authors
This study is relevant to nursing education as it provides evidence to support the use of virtual reality (VR) as a simulation strategy and notes the potential benefits of VR in comparison to the desktop simulation. The manuscript was organized, sections were developed, and followed the scientific article format (Abstract, Introduction, Methods, Results, & Discussion). The Cognitive Affective Model of Immersive Learning (CAMIL) is applicable framework for the study. The unique aspect of the study includes the expansion of the CAMIL framework to include the concept of cognitive absorption. The cited source of evidence contained in the literature review section included 10/47 that were within the last 5 years and were relevant. The sample size was appropriate and was justified by a power analysis. The questionnaire's reliability and validity was referred to through noting of previous research and the researchers provided an overall internal consistency that was adequate. The researchers did not overreach in their interpretation of the results and answered all the hypotheses and purpose of the study. The researchers related the findings to previous research and discussed the significance of the results and it's relationship to the framework. The researchers also provided comments on the significance of the study, addressed limitations, and discussed future research.
My recommendations include:
- lines 41-42 to make the sentence clearer. Identify the developer of the model and that you are using it to frame your study.
- line 225 include the research design
- line 246 include a description of the programming software used to develop the VR simulation, how the participant navigated the environment, and how they picked up and touch objects in the VR environment.
- lines 284-288 Include the percentage of the remaining participants that are not already identified. A table would help with clarification.
- line 312-313, 319,322-323 and 324 - use the actual p value for the three factors ( immersion, curiosity, and control). Add a table with these results.
- Lines 328 -334 address results for H7a, H7b, H7c & H7d found in table 3
- Line 339 there is a discrepancy with the stated diff|=0247 but in table 3 diff|=0248
Comments on the Quality of English Language
Overall the quality of English language was appropriate. There were some grammar and sentence structure concerns.
Author Response
Reviewer 1
This study is relevant to nursing education as it provides evidence to support the use of virtual reality (VR) as a simulation strategy and notes the potential benefits of VR in comparison to the desktop simulation. The manuscript was organized, sections were developed, and followed the scientific article format (Abstract, Introduction, Methods, Results, & Discussion). The Cognitive Affective Model of Immersive Learning (CAMIL) is applicable framework for the study. The unique aspect of the study includes the expansion of the CAMIL framework to include the concept of cognitive absorption. The cited source of evidence contained in the literature review section included 10/47 that were within the last 5 years and were relevant. The sample size was appropriate and was justified by a power analysis. The questionnaire's reliability and validity was referred to through noting of previous research and the researchers provided an overall internal consistency that was adequate. The researchers did not overreach in their interpretation of the results and answered all the hypotheses and purpose of the study. The researchers related the findings to previous research and discussed the significance of the results and it's relationship to the framework. The researchers also provided comments on the significance of the study, addressed limitations, and discussed future research.
Response :
Thank you very much for your positive and encouraging feedback. We truly appreciate your recognition of the study's contribution, methodology, and theoretical framework. We will carefully consider your recommendations and have made revisions where appropriate in the manuscript.
My recommendations include:
Comment 1: lines 41-42 to make the sentence clearer. Identify the developer of the model and that you are using it to frame your study.
Response 1:
Thank you for your suggestion. I have revised the sentence to clarify the developer of the model and its role in framing my study (see line 45-56). As referred to in the following paragraph in the manuscript:
" To address this gap, Makransky and Petersen [13] proposed the Cognitive–Affective Model of Immersive Learning (CAMIL)—a research-based theoretical framework that explains how learning occurs in immersive environments. Although CAMIL is not technology-specific and applies to immersive learning technologies in general, it builds on a recent wave of media comparison studies involving VR. This framework takes a constructivist view of learning, emphasizing the active construction of knowledge through immersive experiences rather than the passive acquisition of information. It provides a structured approach to understanding the complex relationships between technological features, psychological experiences, and learning outcomes in virtual environments. According to CAMIL, the technological features of immersion and interactivity in VR influence presence. Presence influences both affective and cognitive mechanisms, including situational interest, intrinsic motivation, self-efficacy, and cognitive load. These affective and cognitive mechanisms influence learning outcomes."
Comment 2: line 225 include the research design
Response 2:
Thank you for your helpful suggestion. In response, we have revised and expanded the content in Sections 3.1 Conditions for Sample Collection and 3.2 Experimental Content and Research Steps to clearly present the research design, participant recruitment and allocation procedures, and the intervention setting(see Line 243-290).
Comment 3: line 246 include a description of the programming software used to develop the VR simulation, how the participant navigated the environment, and how they picked up and touch objects in the VR environment.
Response 3:
Thank you for your valuable suggestion. I have revised the manuscript to include a detailed description of the software used to develop the VR simulation, as well as how participants navigated and interacted with the virtual environment (see line 279-287). As referred to in the following paragraph in the manuscript:
“The virtual learning environments were built in Unity 3D and programmed in C#. Relevant 3D models were created in 3ds Max and integrated into the system. All con-tent was standardized to ensure consistency between groups, with the only difference being the level of immersion provided by the VR modality. In the IVR group, partici-pants used HTC VIVE Focus Plus head-mounted displays and handheld controllers. They navigated the virtual environment using a joystick and interacted with objects using the controller buttons. The Desktop VR group used a computer, mouse, and key-board. Navigation was performed using arrow keys and mouse movements, while ob-ject interactions were performed using mouse clicks. Participants completed the simu-lation at their own pace”
Comment 4: lines 284-288 Include the percentage of the remaining participants that are not already identified. A table would help with clarification.
Response 4:
Thank you for your insightful suggestion. To address your comment, I have added a table summarizing the demographic characteristics of participants, including the percentages for all subgroups. Please see table 1.
Comment 5: line 312-313, 319,322-323 and 324-use the actual p value for the three factors (immersion, curiosity, and control). Add a table with these results.
Response 5:
Thank you for pointing this out. I have revised the manuscript to report the exact p-values for immersion (p = 0.035) and curiosity (p = 0.006). For the variables with extremely small p-values, including "control" and other predictors, I retained the notation p < .001, which is a common and accepted practice when exact values are not available or are extremely small. Additionally, a summary table of these results has been added to improve clarity (see Table 4).
Comment 6: Lines 328 -334 address results for H7a, H7b, H7c & H7d found in table 3
Response 6:
Thank you for your suggestion. I have added the corresponding statistical results for H7a, H7b, H7c, and H7d in the revised manuscript (see lines 386-396). As referred to in the following paragraph in the manuscript:
“In the multigroup analysis, the permutation test revealed no significant differences in several paths related to presence. Specifically, temporal dissociation did not show a significant difference in its effect on presence (|diff| = 0.044, p = 0.729), leading to the rejection of H7a. Similarly, focused immersion (|diff| = 0.001, p = 0.992) and heightened enjoyment (|diff| = 0.254, p = 0.344) showed no significant differences, resulting in the rejection of H7b and H7c, respectively. Although curiosity yielded a relatively large coefficient difference (|diff| = 0.299), the result was not statistically significant (p = 0.204), and thus, H7d was also rejected. However, the relationship between control and presence showed a significant difference between groups (|diff| = 0.337, p = 0.016), with the IVR group exhibiting a stronger effect. Therefore, H7e was supported (see Table 5).”
Comment 7: Line 339 there is a discrepancy with the stated diff|=0247 but in table 3 diff|=0248
Response 7:
Thank you for pointing this out. This was my mistake. The correct value is 0.248. I have corrected the value in the main text accordingly (see line 396).
Comment 8: Overall the quality of English language was appropriate. There were some grammar and sentence structure concerns.
Response 8:
Thank you for your comment regarding the language quality. We have conducted English language editing through MDPI’s professional editing service (Editing ID: english-92284). We hope the current version meets the language standards required for publication. Thank you again for your valuable feedback.
Reviewer 2 Report
Comments and Suggestions for Authors
This is a fantastic paper. The hypotheses are sound, well evidenced, and insightful. My comments are minor and for polishing.
p3. sentence beginning "Therefore, we posit that when learners..." this sentence is too long and the noun and verb are placed at the end of the sentence instead of the beginning. To make it easier for readers to follow, I recommend swapping sentence structure around and shortening the sentence. i.e. sense of presence should be at the beginning.
terminology - the authors use the terms individuals, samples and subjects. However, these are participants. They chose to participate in your research, they are not subjected to it. I would encourage authors to use this term consistently instead. e.g. "4.1 Sample descriptive statistics
Among the 209 samples collected, 79 (37.8%) were 41–50 years old, and 60 (28.7%) 284
were 51–60 years old." could be "Participant characteristics. Among the 209 participants, 79 (37.8%) were 41–50 years old, and 60 (28.7%) were 51–60 years old."
I particularly enjoyed your discussion, insightful and practical. thanks!
Author Response
Reviewer 2
This is a fantastic paper. The hypotheses are sound, well evidenced, and insightful. My comments are minor and for polishing.
Thank you very much for your positive and encouraging feedback. We truly appreciate your kind words regarding the quality of our work, as well as your helpful suggestions for further polishing the manuscript. We have carefully addressed your minor comments and revised the manuscript accordingly.
Comment 1: p3. sentence beginning "Therefore, we posit that when learners..." this sentence is too long and the noun and verb are placed at the end of the sentence instead of the beginning. To make it easier for readers to follow, I recommend swapping sentence structure around and shortening the sentence. i.e. sense of presence should be at the beginning.
Response 1:
Thank you for your kind feedback and valuable suggestion. We revised the sentence on page 3. As referred to in the following paragraph in the manuscript:
“Therefore, we posit that learners’ sense of presence in the virtual environment may be enhanced through several experiential aspects of VR interaction. These include the following: experiencing temporal dissociation (becoming so engaged that they lose track of time); achieving focused immersion (concentrating on the task while ignoring external distractions); feeling heightened enjoyment during the learning process; exerting control by consciously mastering task execution; and engaging curiosity, which drives imaginative involvement throughout the experience. Based on these relationships, we propose the following hypothesis” (see lines 111-118).
Comment 2: terminology - the authors use the terms individuals, samples and subjects. However, these are participants. They chose to participate in your research, they are not subjected to it. I would encourage authors to use this term consistently instead. e.g., "4.1 Sample descriptive statistics Among the 209 samples collected, 79 (37.8%) were 41–50 years old, and 60 (28.7%) 284 were 51–60 years old." could be "Participant characteristics. Among the 209 participants, 79 (37.8%) were 41–50 years old, and 60 (28.7%) were 51–60 years old."
Response 2:
Thank you for your valuable suggestion. In response, I have revised the manuscript to consistently use the term “participants” in place of “individuals,” “samples,” and “subjects” when referring to those who took part in the study.
Comment 3: I particularly enjoyed your discussion, insightful and practical. thanks!
Response 3:
Thank you very much for your kind words. We are glad to hear that you found the discussion insightful and practical. Your feedback is greatly appreciated.
Reviewer 3 Report
Comments and Suggestions for Authors
Thank you for giving me the opportunity to read this interesting manuscript. The topic is interesting; there are, however, there are several methodological limitations that need to be addressed before this paper can be considered to be published.
It is very positive that the authors present limitations in the manuscript; there is, however, lacking reflections on the study design and methodological considerations.
There is no specific rationale on which authors ground their introduction, aim and study. The background section is too general, and there is a need to include more of what the CAMIL model consists of specifically. There is also a need to include how the VR intervention aligns with the CAMIL model. The author should consider including the background information in the introduction section, and putting the hypothesis sections in the methodology.
The manuscript lacks pivotal descriptions of the methodology and design approach, specifically to study design, recruitment process, intervention setting and a clear description of the used instrument. A couple of reflections: What was the rationale for comparing desktop simulation versus VR? Would it be more relevant to compare real-life skill training with VR? There is a need for more comprehensive information on the instrument used in this study.
Comments on the Quality of English Language
There are many grammatical errors throughout the manuscript, which makes it difficult to follow the authors’ rationale. I recommend the authors to use a professional language editing and proofreading service.
Author Response
Reviewer 3
Thank you for giving me the opportunity to read this interesting manuscript. The topic is interesting; there are, however, there are several methodological limitations that need to be addressed before this paper can be considered to be published.
Thank you very much for your thoughtful and constructive feedback. We appreciate your recognition of the topic's relevance and your careful review of the manuscript. Detailed responses to each of your comments are provided below. We hope that the revised version meets your expectations and improves the overall quality of the study.
Comment 1: It is very positive that the authors present limitations in the manuscript; there is, however, lacking reflections on the study design and methodological considerations.
Response 1:
Thank you for your valuable comment. We appreciate your positive feedback regarding our discussion of study limitations. In response, we have revised and expanded the content in Sections 3.1 Conditions for Sample Collection and 3.2 Experimental Content and Research Steps to clearly present the research design, participant recruitment and allocation procedures, and the intervention setting (see Line 242-290). We also have revised Section 5.3 (Limitations and suggestions) to include additional reflections on the study design and methodological considerations (see Line 496-503).
Comment 2: There is no specific rationale on which authors ground their introduction, aim and study. The background section is too general, and there is a need to include more of what the CAMIL model consists of specifically. There is also a need to include how the VR intervention aligns with the CAMIL model. The author should consider including the background information in the introduction section, and putting the hypothesis sections in the methodology.
Response 2:
Thank you for your valuable comment. In response, we revised the Introduction (Lines 44–56) to provide a clearer and more specific explanation of the CAMIL. To address the comment regarding the alignment between the VR intervention and the CAMIL model, we have expanded Section 3.2 (Experimental Content and Research Steps) to explicitly describe how the instructional content was structured to reflect the theoretical constructs of CAMIL (Lines 261–270).
Regarding the suggestion to relocate the hypotheses to the methodology section, we appreciate your perspective but have chosen to keep the hypotheses within Section 2 (Theoretical Background and Hypotheses). This section presents the conceptual development based on literature and theoretical reasoning, which directly leads to the proposed hypotheses. We believe this placement provides a more coherent narrative linking theory to research design.
Comment 3: The manuscript lacks pivotal descriptions of the methodology and design approach, specifically to study design, recruitment process, intervention setting and a clear description of the used instrument. A couple of reflections: What was the rationale for comparing desktop simulation versus VR? Would it be more relevant to compare real-life skill training with VR? There is a need for more comprehensive information on the instrument used in this study.
Thank you for your valuable and detailed comments. In response, we have revised several sections of the manuscript to clarify the methodology and rationale for our design choices.
Response 3:
First, we have expanded Section 3.1 (Conditions for Sample Collection) and Section 3.2 (Experimental Content and Research Steps) to more clearly describe the study design (randomized experimental design), recruitment process (inclusion and exclusion criteria, random assignment, sample size calculation), and the intervention setting for both immersive VR (IVR) and Desktop VR groups (see Lines 242–290).
Second, regarding the rationale for comparing Desktop VR instead of real-life skill training, we have clarified this in the revised Introduction (Lines 87–92). As noted, existing studies have primarily compared VR with traditional or video-based instruction. However, our study aimed to isolate the effect of immersion level, which is a core construct in the CAMIL framework. By comparing IVR and Desktop VR—both using identical instructional content, system logic, and task sequence—we were able to control for instructional variables and attribute differences in learning outcomes to the degree of immersion.
Third, we have revised and expanded Section 3.4 (Measurement Tools) to provide a more comprehensive description of the instrument used in this study. We now include information on the number of items, sources of each construct, sample items, scale format, and internal consistency (Cronbach’s α).
Comment 4: There are many grammatical errors throughout the manuscript, which makes it difficult to follow the authors’ rationale. I recommend the authors to use a professional language editing and proofreading service.
Response 4:
Thank you for your comment regarding the language quality. We have conducted English language editing through MDPI’s professional editing service (Editing ID: english-92284). We hope the current version meets the language standards required for publication. Thank you again for your valuable feedback.
Round 2
Reviewer 3 Report
Comments and Suggestions for Authors
Thank you for the revised manuscript. This draft meets the scientific criteria for publication.
Author Response
Comment 1: Thank you for the revised manuscript. This draft meets the scientific criteria for publication.
Response 1:
I sincerely appreciate your thorough review and constructive comments throughout the revision process.